

# Many-body localization in the Fock space of natural orbitals

**Wouter Buijsman**[1*]**, Vladimir Gritsev**[1, 2]**, Vadim Cheianov**[3]

**1** Institute for Theoretical Physics and Delta Institute for Theoretical Physics, University of Amsterdam, Science Park 904, 1098 XH Amsterdam, The Netherlands
**2** Russian Quantum Center, Skolkovo, Moscow 143025, Russia
**3** Instituut-Lorentz and Delta Institute for Theoretical Physics, Universiteit Leiden, P.O. Box 9506, 2300 RA Leiden, The Netherlands

* w.buijsman@uva.nl

## Abstract

We study the eigenstates of a paradigmatic model of many-body localization in the Fock basis constructed out of the natural orbitals. By numerically studying the participation ratio, we identify a sharp crossover between different phases at a disorder strength close to the disorder strength at which subdiffusive behaviour sets in, significantly below the many-body localization transition. We repeat the analysis in the conventionally used computational basis, and show that many-body localized eigenstates are much stronger localized in the Fock basis constructed out of the natural orbitals than in the computational basis.

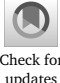
## 1 Introduction

Localization of many-body states in the Fock space [1], a phenomenon referred to as many-body localization (MBL), has become a trending research field during the last decade [2, 3].

Inspired by the seminal work of Basko, Aleiner and Althshuler [4], a large number of investigations has revealed various intriguing properties of the many-body localized phase, among them the persistence up to infinite temperature [5], the separation from the thermal phase by a phase transition [6,7], and the growth of entanglement in the absence of transport [8,9]. The interest for MBL is mainly driven by the notion that many-body localized systems violate the fundamental assumption of statistical mechanics that a non-integrable system can serve as its own heath bath, a phenomenon that has been near-rigorously proven to exist only recently [10].

Over the last few years, it has become clear [11] that not only the many-body localized phase, but also the thermal phase in the vicinity of the MBL transition ('critical phase') displays remarkable properties [12], such as subdiffusion [13,14], subthermal entanglement scaling [15], bimodality of the entanglement entropy distribution [16,17], and the violation of the eigenstate thermalization hypothesis [18,19]. The latter can be deduced from the violation of the Berry conjecture [20], roughly stating that the eigenstates of thermal systems are spread out over the full Hilbert space in any local basis. In this work, we study the spreading of eigenstates over the Hilbert space for a paradigmatic model of many-body localization. By numerically studying the participation ratio for a finite-size system, we identify a sharp crossover between different phases at a disorder strength close to the disorder strength at which subdiffusive behaviour [21] and the departure from Poissonian level statistics [7] sets in.

We identify the crossover in the Fock basis constructed out of the natural orbitals, and repeat the analysis in the conventionally used computational basis. The natural orbitals and their corresponding occupation numbers resulting from the diagonalization of the one-particle density matrix [22] recently gained significant attention in the field of MBL [23–27]. It was found [23] that the occupation numbers exhibit qualitatively different statistics in the thermal and the many-body localized phase, allowing them to be used as a probe for the MBL transition [7,28]. Based on these statistics, we argue that the scope can be naturally broadened by studying MBL in the Fock basis constructed out of the natural orbitals. We show that many-body localized eigenstates are much stronger localized in this basis than in the computational basis, and state how studying MBL in this basis might lead to a better understanding of the many-body localized phase.

## 2 The model

We consider the standard model of MBL, a 1-dimensional chain of spinless fermions with nearest-neighbor interactions and random onsite disorder. The Hamiltonian $H$ reads

$$H = \frac{1}{2}\sum_{i=1}^{L}\left(c_i^{\dagger}c_{i+1} + c_i c_{i+1}^{\dagger}\right) + \sum_{i=1}^{L} h_i\left(n_i - \frac{1}{2}\right) + \Delta\sum_{i=1}^{L}\left(n_i - \frac{1}{2}\right)\left(n_{i+1} - \frac{1}{2}\right), \qquad (1)$$

with $n_i = c_i^{\dagger}c_i$, where $\{c_i^{\dagger}, c_j\} = \delta_{i,j}$ in units $\hbar = 1$. This model is equivalent to a disordered spin-1/2 Heisenberg chain via a Jordan-Wigner transformation. In what follows, periodic boundary conditions $c_{i+L} \equiv c_i$ have been imposed, and the number of fermions is set to $L/2$ (half-filling) with $L$ ranging from 10 to 16. For consistency with previous works [3], we sample the onsite disorder $h_i$ from a uniform distribution ranging over $[-W, W]$, and set $\Delta = 1$. We generate ensemble averages from 1000 disorder realizations, and for each disorder realization we only consider the eigenstate with the energy closest to the middle $(\min(E) + \max(E))/2$ of the spectrum $\{E_i\}$. For these parameters, the model is believed to exhibit an MBL transition at $W \approx 3.6$ [7,29].

## 3 Fock space of natural orbitals

The one-particle density matrix (OPDM) $\rho$ of an eigenstate $|\Psi\rangle$ is element-wise defined [22] as $\rho_{ij} = \langle\Psi|c_i^\dagger c_j|\Psi\rangle$. Diagonalizing $\rho$ by solving

$$\rho|\phi_i\rangle = n_i|\phi_i\rangle \tag{2}$$

gives the occupation numbers $0 \le n_i \le 1$ and the corresponding natural orbitals $|\phi_i\rangle$. In the non-interacting case $\Delta = 0$, the eigenstates of Hamiltonian (1) are given by exterior products of natural orbitals, known as Slater determinants. These Slater determinants are characterized by occupation numbers $n_i = 1$ and $n_i = 0$ for the occupied and unoccupied natural orbitals, respectively. In the language of second quantization, they are created from the vacuum $|0\rangle$ as

$$|\Psi\rangle = \left(\prod_{\{i|n_i=1\}} d_i^\dagger\right)|0\rangle, \qquad d_i^\dagger = \sum_{j=1}^{L}\phi_i(j)c_j^\dagger, \tag{3}$$

where $\phi_i(j)$ is the $j$-th element of $\phi_i$.

For many-body localized eigenstates (*i.e.* returning to $\Delta = 1$), it was argued and validated numerically recently [23] that the ensemble average of the occupation discontinuity $\Delta n \in [0,1]$ given by

$$\Delta n = \max_i(n_i - n_{i+1}), \tag{4}$$

with $\{n_i\}$ sorted in descending order can be used as a probe for the MBL transition, being given by $\langle\Delta n\rangle \approx 1$ in the many-body localized and $\langle\Delta n\rangle$ significantly smaller than 1 in the thermal phase [23, 25]. This observation initiated studies on various aspects of OPDMs [24–27, 30] of many-body localized eigenstates. The characterization $\langle\Delta n\rangle \approx 1$ is reminiscent of Anderson localization, where states are characterized by $\Delta n = 1$. Based on this, one might expect that many-body localized eigenstates can be well approximated by the single Slater determinant constructed out of the heighest occupied natural orbitals. This Slater determinant can be seen as a basis state of the Fock space of Slater determinants constructed out of the natural orbitals, for which the basis states are created by applying subsets of $\{d_1^\dagger, d_2^\dagger, \ldots, d_L^\dagger\}$ on $|0\rangle$. Going further, one might hypothesize that many-body localized eigenstates are strongly localized in the Fock space constructed out of the natural orbitals.

Here, we we aim to validate the above hypothesis. Let $|\Psi^{(0)}\rangle$ denote the Slater determinant constructed out of the $N$ highest occupied natural orbitals of an $N$-body eigenstate $|\Psi\rangle$, and let $\{|\Psi_i^{(n)}\rangle\}$ with index $i$ denote the sets of Slater determinants having $n$ particle-hole excitations compared to $|\Psi^{(0)}\rangle$. The elements of $\{|\Psi_i^{(n)}\rangle\}$ with index i are created by applying $n$ creation and $n$ annihilation operators, all with distinct indices, of natural orbitals on $|\Psi^{(0)}\rangle$. Supposing $\{n_i\}$ is sorted in descending order, the indices $\{1, 2, \ldots, N\}$ label the $N$ highest and the indices $\{N+1, N+2, \ldots, L\}$ label the $L-N$ lowest occupied natural orbitals. Explicitly, then

$$|\Psi_0\rangle = d_1^\dagger \ldots d_N^\dagger|0\rangle, \tag{5}$$

while a state $|\Psi_i^{(n)}\rangle$ is constructed as

$$|\Psi_i^{(n)}\rangle = \underbrace{d_{j_1}^\dagger \cdots d_{j_n}^\dagger}_{\{j_1,\ldots,j_n\}>N}\underbrace{d_{k_1}\cdots d_{k_n}}_{\{k_1,\ldots,k_n\}\le N}|\Psi_0\rangle, \tag{6}$$

with the indices $j_1, \ldots j_n$ and $k_1, \ldots, k_n$ all distinct.

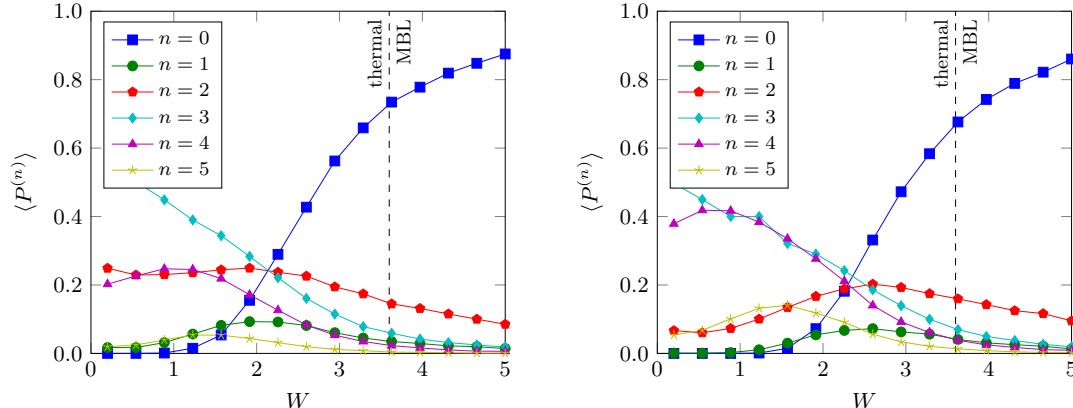

Figure 1: Ensemble averages of $P^{(n)}$ as given in eq. (7) for $n = 0, 1, 2, 3, 4, 5$ at $L = 14$ (left) and $L = 16$ (right). The MBL transition is indicated by a dashed line. Averages are determined from 1000 distinct eigenstates.

The full Fock basis is spanned by $\{|\Psi_i^{(n)}\rangle\}$ with indices $i$ and $n$. A particularly simple way to study the structure of eigenstates in the Fock basis constructed out of the natural orbitals is provided by the quantity

$$P^{(n)} = \sum_i |\langle \Psi_i^{(n)} | \Psi \rangle|^2, \tag{7}$$

which gives the distribution of $|\Psi\rangle$ over basis states with a given number $n$ of particle-hole excitations compared to $|\Psi^{(0)}\rangle$. A fully localized eigenstate is characterized by $P^{(0)} = 1$ and $P^{(n)} = 0$ for $n \geq 1$, while $P^{(n)} \propto \dim\left(\{|\Psi_i^{(n)}\rangle\}\right)$ with the proportionality factor chosen such that $\sum_n P^{(n)} = 1$ if $|\Psi\rangle$ is an uniform superposition of all basis states.

Figure 1 shows the ensemble average of $P^{(n)}$ for several values of $n$ as a function of the disorder strength $W$. Many-body localized eigenstates are well localized in the Fock basis of natural orbitals. On average, eigenstates are mainly composed out of basis states with low values of $n$, which is consistent with the interpretation of MBL as localization of many-body states in the Fock space [4]. No clear signatures of the MBL transition can be observed, and on average eigenstates seem to remain localized at disorder strengths even below the MBL transition. This is consistent with a previous investigation [18] on thermalization of eigenstates from the point of the Berry conjecture [20] indicating the violation of the eigenstate thermalization hypothesis [19] at $W = 1.6$. As a matter of fact, the single-particle states $\phi_i$ are known to be well-localized in the MBL phase, while they are far more extended in the delocalized phase [23]. We observe $\langle P^{(2)} \rangle \gg \langle P^{(1)} \rangle$ in the many-body localized phase, which we expect to be a consequence of the basis transformation $c_i \to d_i$ from the computational to the Fock basis characterized by $\langle \Psi | d_i^\dagger d_j | \Psi \rangle = 0$ for $i \neq j$.

## 4 Probing crossovers

In this Section, we show the existence of a crossover between different phases at a disorder strength close to the disorder strength at which subdiffusive behaviour [21] and the departure from Poissonian level statistics [7] sets in, and identify it to be sharp. We do this by studying the participation ratio

$$\text{PR} = \frac{1}{\sum_{i,n} |\langle \Psi_i^{(n)} | \Psi \rangle|^4} \tag{8}$$

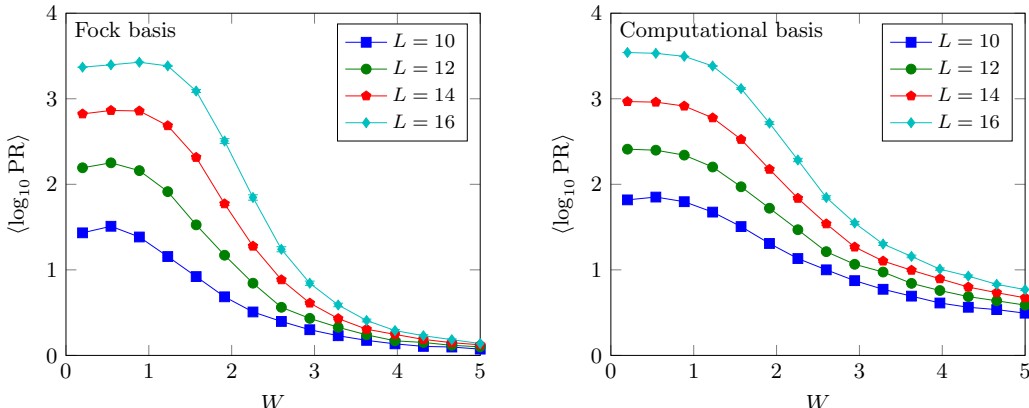

Figure 2: The ensemble averages of $\log_{10} \mathrm{PR}$ as given in eq. (8) for $L = 10, 12, 14, 16$ in the Fock basis (left) and the compuational basis (right). Averages are taken over 1000 eigenstates. Error bars (mostly smaller than the marker size) are determined by jackknife resampling.

in both the Fock basis introduced in Section 3 and the conventionally used computational basis of Hamiltonian (1). When considering the computational basis, the summation over the indices $i$ and $n$ should be read as a summation over the indices of all basis states. Note that, since we aim to investigate a crossover in a specific basis, dynamical [31–33] or basis-independent probes based on *e.g.* level statistics [7, 34] or entanglement [7] can not be used. The MBL transition has been identified from the participation ratio in the computational basis in a previous study [35].

For a fully localized eigenstate, $\mathrm{PR} = 1$, while $\mathrm{PR} = N$ if $|\Psi\rangle$ is a uniform superposition of $N$ basis states. Thus, for a given basis, PR can be interpreted as a measure of the effective Hilbert space dimension in which an eigenstate is confined. For $L = 16$, the participation ratio varies over roughly 4 orders of magnitude when going from a fully thermal to a fully localized eigenstate. To account for this, we here focus on the logarithm $\log_{10} \mathrm{PR}$. We study successively (a) the ensemble average, (b) the variance within the ensemble, (c) the scaling with the Hilbert space dimension and (d) the histograms as a function of the disorder strength $W$. We have verified that focusing on PR instead of $\log_{10} \mathrm{PR}$ does not qualitatively alter our conclusions.

**Ensemble average**    First, we study the ensemble average of $\log_{10} \mathrm{PR}$. Figure 2 shows $\langle \log_{10} \mathrm{PR} \rangle$ for system sizes $L = 10, 12, 14, 16$ as a function of $W$ in both the Fock and computational basis. One observes a disorder strength-dependency for $W \gtrsim 1.7$ at $L = 16$ in both bases, suggesting the presence of a crossover from a phase with thermal to a phase with non-thermal eigenstates starting at $W \approx 1.7$. We observe that $\langle \log_{10} \mathrm{PR} \rangle$ is significantly lower in the Fock basis compared to the computational basis for $W \gtrsim 1.7$ for all system sizes, indicating much stronger localization in the former compared to the latter.

**Ensemble variance**    Second, we study the variance of $\log_{10} \mathrm{PR}$ within the ensemble, given by

$$\mathrm{var}(\log_{10} \mathrm{PR}) = \langle (\log_{10} \mathrm{PR} - \langle \log_{10} \mathrm{PR} \rangle)^2 \rangle. \tag{9}$$

For $L \to \infty$, this quantity is expected to vanish in a strongly delocalized and a strongly localized phase, and to peak at a crossover due to the mixture and coexistence of thermal and non-thermal eigenstates within the ensemble [36]. This idea has been applied to probe the MBL transition previously [23]. Noteworthy, observations pointing towards similar conclusions as drawn in this Section have been obtained by studying the ensemble variance of the

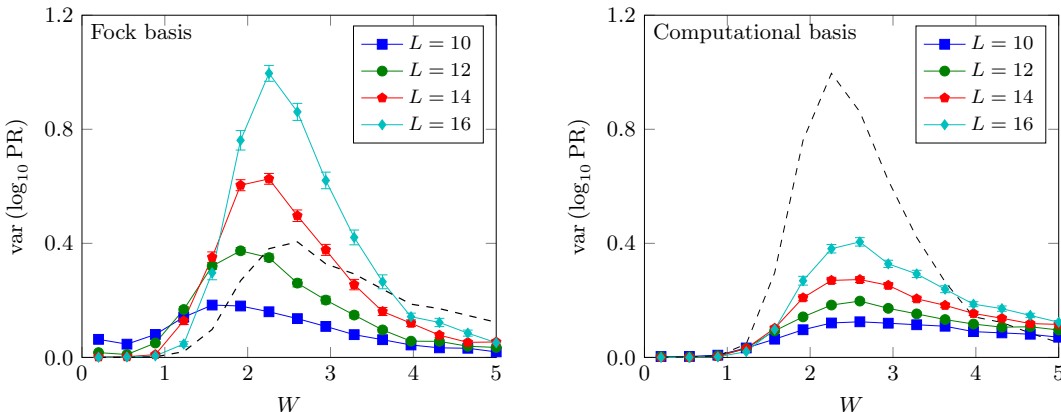

Figure 3: The ensemble variance $\text{var}\big(\log_{10} \text{PR}\big)$ as given in eq. (9) for $L = 10, 12, 14, 16$ in the Fock basis (left) and the compuational basis (right). Averages are taken over 1000 eigenstates. Error bars (mostly smaller than the marker size) are determined by jackknife resampling. For comparison, the dashed lines in the left (right) plot indicate $\text{var}\big(\log_{10} \text{PR}\big)$ for $L = 16$ in the Fock (computational) basis.

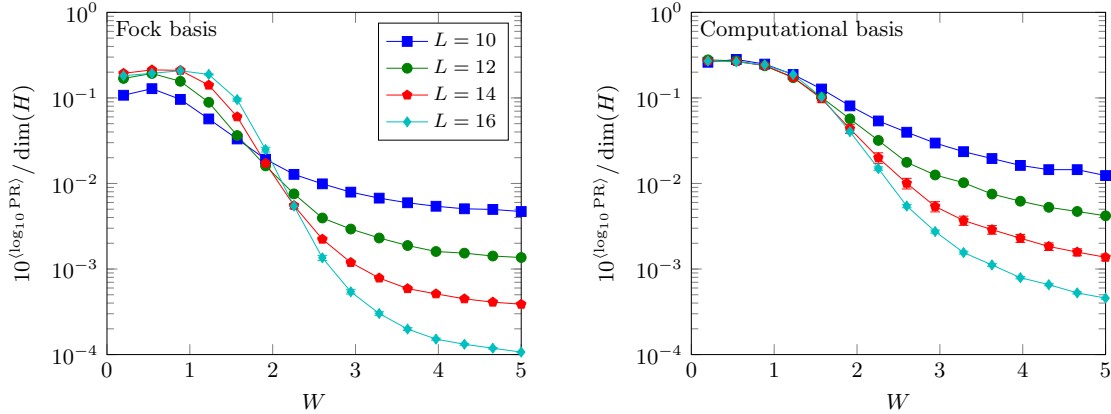

Figure 4: Plots of $10^{\langle \log_{10} \text{PR} \rangle} / \dim(H)$ as given in eq. (8) for $L = 10, 12, 14, 16$ in the Fock basis (left) and the compuational basis (right). Averages are taken over 1000 eigenstates. Error bars (mostly smaller than the marker size) are determined by jackknife resampling.

bipartite entanglement entropy [16,17]. Figure 3 show $\text{var}(\log_{10} \text{PR})$ in the Fock and the computational bases for $L = 10, 12, 14, 16$ as a function of $W$. In both bases, one observes a peak at $W \approx 2.3$ for $L = 16$, thereby supporting the interpretation of Figure 2. For the system sizes under consideration, the peak becomes increasingly sharper with increasing system size. Interestingly, the crossover is close to the disorder strength at which subdiffusive behaviour [21] and the departure from Poissonian level statistics [7] sets in. It is an open question is if the peak moves towards the MBL transition at $W \approx 3.6$ in the thermodynamic limit $L \to \infty$.

**System size scaling** Third, we study the scaling of the participation ratio with the Hilbert space dimension when varying $L$. As mentioned above, PR can be interpreted as a measure for the dimensionality of the effective Hilbert space in which an eigenstate is confined. Hence, $10^{\langle \log_{10} \text{PR} \rangle} / \dim(H)$ can be seen as a measure for the fraction of the full Hilbert space that is occupied by an eigenstate on average. Figure 4 shows the ensemble average of the above quantity in the Fock and computational bases for $L = 10, 12, 14, 16$. Here, $\dim(H)$ is the dimension of Hamiltonian (1) with the focus restricted to the sector with $L/2$ fermions, which

scales exponentially with $L$ up to good approximation. Again, the figure suggests a crossover at $W \approx 2.3$ in both bases, even though the effect is significantly less pronounced in the computational basis.

**Inspection of the histograms**    Finally, we perform a visual inspection of the histograms of $\log_{10} \mathrm{PR}$ at disorder strengths around $W \approx 2.3$. Figure 5 shows histograms of $\log_{10} \mathrm{PR}$ determined in both the Fock basis and the computational basis for $L = 16$ at several disorder strengths ranging from $W \approx 1.6$ to $W \approx 3.3$. Focusing on the Fock basis, one observes a qual-

Figure 5: Normalized histograms of $\log_{10} \mathrm{PR}$ determined in the Fock basis (solid lines, filled) and the computational basis (dashed lines, unfilled) for $L = 16$ at several disorder strengths ranging from $W \approx 1.6$ to $W \approx 3.3$. Each histogram consists of 1000 entries.

itative difference in the structure of eigenstates when comparing the histograms for $W \approx 1.6$ and $W \approx 3.2$. A similar effect can be observed when focusing on the computational basis, even though the effect is significantly less clear in that case.

## 5  Discussion and conclusions

In this work, we have studied many-body localization in the Fock basis constructed out of the natural orbitals. Focusing on the participation ratio as given in eq. (8) for Hamiltonian (1), we have shown that many-body localized eigenstates are strongly localized in this basis, in fact more strongly than in the typically used computational basis. We expect that future studies in this basis might reveal new or quantitatively more accurate descriptions of the many-body localized phase. In particular, working in this basis might lead to a better understanding of the multifractality observed in the many-body localized phase [7, 37] by focusing on *e.g.* the basis-dependent participation entropy [38].

When considering $P^{(0)}$ as given in eq. (7) as a measure of the localization of an eigenstate, one can not exclude that different single-particle states leading to even more strongly localized eigenstates can be found [39]. An iterative algorithm to find the optimal single-particle states archieving this has been proposed [40]. However, convergence of this algorithm is not guaranteed. We hope this work can initiate a search for even more optimal bases in which to study MBL, potentially leading to more stringent conclusions on the crossover in eigenstate statistics observed in this work.

By studying the participation ratio as given in eq. (8), we have identified a sharp crossover between different phases at a disorder strength close to the disorder strength at which subdiffusive behaviour [21] and the departure from Poissonian level statistics [7] sets in, significantly below the MBL transition [7]. Further investigations on the relation between these different phenomena might be valuable, in particular in the thermodynamic limit $L \rightarrow \infty$, where the departure from Poissonian level statistics with disorder strength is expected to coincide [7] with the MBL transition, and where the subdiffusive phase is suggested to be absent [41].

## Acknowledgements

This work is part of the Delta-ITP consortium, a program of the Netherlands Organization for Scientific Research (NWO) that is funded by the Dutch Ministry of Education, Culture and Science (OCW).

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
