# Peer review of "Many-body localization in the Fock space of natural orbitals"

_SciPost Physics, doi:SciPost Phys. 4, 038 (2018)_

## Round 1 · Referee Report · Anonymous · 2018-2-6

Strengths

1- Interesting and timely subject
2- interesting approach to Fock space localization

Weaknesses

1- statistical analysis can be improved

Report

The authors present interesting results of localization of
eigenstates in an MBL system in the basis of natural orbitals of the
one particle density matrix. They find that eigenstates are
significantly more localized in this Fock basis compared to the
conventional computational basis and claim that there is an onset of
localization much below the critical disorder strength of the MBL
transition, in the region of the phase diagram, where subdiffusive
transport was observed.

The paper is overall quite readable and accessible and addresses an
interesting and important aspect of many-body localization.
Nevertheless, I think that several aspects of the analysis of the
results should be improved significantly before publication of this
work. Since the study of many-body localization is in practice
confined to very small system sizes, it is of tremendous importance to
analyze the effects of system sizes and statistical fluctuations.
Therefore, I suggest that the authors consider the following
improvements, which could add significant value to the paper:

1) The authors perform an analysis of statistical errors over the
ensemble of 1000 single central eigenstates obtained from independent
disorder realizations. For this, they split their ensemble in two
halves and take the difference in the results of the two subsamples as
an estimate for the uncertainty. While this procedure yields a very
rough estimate of the error, it suffers from large uncertainties in
the error estimate and also potentially overestimates the uncertainty,
since the underlying amount of data relies only on half samples.
I suggest that the authors change their analysis to use standard
resampling methods such as either the jackknife or bootstrap method to
obtain more reliable standard error estimates. The such obtained
errorbars should be attached in all figures.
I expect that these error estimates may explain the nonmonotonic
behavior of the PR and in particular var(PR) in Figs 2 and 3, most
pronounced for L=16.

2) The analysis of the evolution with system sizes should be done in a
more systematic way. In particular I think that it would be very
interesting to use the data from Fig. 2 and extract the scaling with
system size in the two bases. Apparently, the PR grows exponentially
with the system size, and one could estimate the corresponding scaling
exponent by fitting an ansatz of the form PR(L) = a*exp(d*L). The
exponents d appear to be slightly different in the two bases and it
looks like d roughly goes to zero at the MBL transition. In the
computational basis, the scaling of log(PR) in the MBL phase has been
reported to show a logarithmic growth with system size, which might be
a difference to the behavior in the Fock basis.

3) It is interesting but not surprising to see that also the PR
in the Fock basis shows a peak in its variance in the ergodic phase
close to the MBL transition. This is probably for the same reason as
the peak in the eigenstate entanglement entropy variance observed in
the same parameter regime. If the authors could present the full
histograms of the PR in the Fock basis, it would be interesting to
check how the distribution looks like at different disorder strengths.

4) If the behavior suggested in 3) is indeed observed, it would be
fully consistent with the previously observed phenomenology in the
distributions of the entanglement entropy, which exhibit bimodal
distributions close to the critical point, pointing to a mixture of
localized and delocalized states.

Overall, I think that this work touches an interesting topic but can
be improved significantly in terms of a systematic analysis of the
data.

Requested changes

1- Replace error analysis by standard bootstrap or jackknife methods
2- Present scaling with system size
3- Show full distributions of IPR
4- Add errorbars to all graphs, indicate number of samples

  • validity: good
  • significance: good
  • originality: high
  • clarity: high
  • formatting: excellent
  • grammar: excellent

Author:  Wouter Buijsman  on 2018-04-12  [id 239]

(in reply to Report 1 on 2018-02-06)

Errors in user-supplied markup (flagged; corrections coming soon)

We would like to thank the referee for the careful reading of the manuscript and for bringing up constructive comments. A revised version (v2) of the manuscript has been submitted. Please see our replies below.

$\textbf{Point 1}$
==================================================
The authors perform an analysis of statistical errors over the ensemble of 1000 single central eigenstates obtained from independent disorder realizations. For this, they split their ensemble in two halves and take the
difference in the results of the two subsamples as an estimate for the uncertainty. While this procedure yields a
very rough estimate of the error, it suffers from large uncertainties in the error estimate and also potentially
overestimates the uncertainty, since the underlying amount of data relies only on half samples. I suggest that the
authors change their analysis to use standard resampling methods such as either the jackknife or bootstrap
method to obtain more reliable standard error estimates. The such obtained errorbars should be attached in all figures. I expect that these error estimates may explain the nonmonotonic behavior of the PR and in particular var(PR) in Figs 2 and 3, most pronounced for L=16.
==================================================

In the revised manuscript, we have added error bars to all plots except the one in the first figure (justified below). The errors are estimated by using the jackknife resampling method. Also, we have indicated the number of samples in the captions of the figures.

An error estimate only has a clear interpretation if the sampled data roughly follows a Gaussian distribution. In view of this, we have shifted the focus from the participation ratio to the logarithm of the participation ratio. We have verified that this does not qualitatively alter any of the results. Figure 1 shows averages taken from various qualitatively different distributions, for which we were not able to find an error estimate with a clear interpretation. As the figure serves only an illustrative purpose, we think this is not a major issue.

$\textbf{Point 2}$
==================================================
The analysis of the evolution with system sizes should be done in a more systematic way. In particular I think that it would be very interesting to use the data from Fig. 2 and extract the scaling with system size in the two bases. Apparently, the PR grows exponentially with the system size, and one could estimate the corresponding scaling exponent by fitting an ansatz of the form PR(L) = a*exp(d*L). The exponents d appear to be slightly different in the two bases and it looks like d roughly goes to zero at the MBL transition. In the computational basis, the scaling of log(PR) in the MBL phase has been reported to show a logarithmic growth with system size, which might be a difference to the behavior in the Fock basis.
==================================================

In the revised manuscript, we have included an analysis of the system size scaling of the data shown in Figure 2 of the revised manuscript. Due to the limited range of system sizes and the rather fuzzy system-size dependence over a wide range of disorder strengths, we were unfortunately not able to obtain reliable estimates of the corresponding scaling coefficients.

$\textbf{Point 3}$
==================================================
It is interesting but not surprising to see that also the PR in the Fock basis shows a peak in its variance in the ergodic phase close to the MBL transition. This is probably for the same reason as the peak in the eigenstate entanglement entropy variance observed in the same parameter regime. If the authors could present the full histograms of the $\text{PR}$ in the Fock basis, it would be interesting to check how the distribution looks like at different disorder strengths.
==================================================

We have included histograms of the logarithm of $\text{PR}$ for several disorder strengths in Figure 5 of the revised manuscript.

$\textbf{Point 4}$
==================================================
If the behavior suggested in 3) is indeed observed, it would be fully consistent with the previously observed phenomenology in the distributions of the entanglement entropy, which exhibit bimodal distributions close to the critical point, pointing to a mixture of localized and delocalized states.
==================================================

We would like to thank the referee for bringing this to our attention. In the revised manuscript, we have added references to Phys. Rev. B 94, 184202 (2016) and Phys. Rev. X 7, 021013 (2017) in the introduction and in Section 3.

---

## Round 1 · Referee Report · Anonymous · 2018-3-15

Strengths

1. This work studies the participation ratio of eigenstates in the middle of the many-body spectrum of the random field Heisenberg model as a function of disorder strength. The quantity is studied in the Fock space basis of eigenstates of the one-particle density matrix. The main result of this is that the eigenstates appear much strongly localized in this basis in comparison to the computational basis which is conventionally used to study the participation ratio of the eigenstates. Due to this the authors conclude that the location of the critical point shifts to lower disorder strength.

Weaknesses

1. The first part of the paper describing the basis idea is clearly written. The construction of the Fock state basis using the natural orbitals could be described in more detail. The interpretation of the data in Fig. 1 is a bit unclear. For n=0, it is clear that the participation is tending to 1 in the localized phase. Why doesn’t the same effect occur for n>0? Why does n=1 curve for P^{(n)} appear qualitatively different from the n=0 curve? Even in the thermal phase n=3,4 curves are around 0.5. I would have expected to be smaller. The natural orbitals presumably in the thermal phase are completely delocalized. Is that true? Since, this is an important aspect of the work and probably the subsequent analysis depends crucially on the behaviour of this quantity, it would be suitable if the the data and its interpretation is clarified substantially. Presumably, further clarity can be achieved by studying the finite size scaling of this quantity.

2. The average PR shown in fig. 2 shows the two different regimes. The exponential growth with system size at low disorder in the thermal and the convergence to 1 for all system sizes clearly show the two regimes quite clearly. The authors use this to conclude that the MBL regime sets in at lower disorder. This is likely correct but the analysis used to reach this conclusion is a bit loose. Scaling the data to extract the critical disorder strength would more thoroughly demonstrate this point. The authors should do this for further clarity.

3. The authors also use the variance of the participation to find the crossover regime. For system sizes 10 and 12, the peak is imperceptible but becomes stronger for system sizes 14 and 16. The delocalised eigenstates satisfy the eigenstate thermalization hypothesis. What is the scaling of the participation ratio expected in the thermal phase based on that? By using the appropriate scaling the location of the critical regime can be better estimated from this quantity.

Report

The results presented in this paper are not unexpected, it is nonetheless an interesting observation and warrants publication.

Requested changes

1. The analysis and description of the numerical results can be sharpened for further clarification.

2. The data presented in the plots have no error bars. A careful error analysis for disordered systems is crucial.

3. I would recommend the manuscript for publication with the above mentioned changes.

  • validity: ok
  • significance: ok
  • originality: low
  • clarity: ok
  • formatting: reasonable
  • grammar: good

Author:  Wouter Buijsman  on 2018-04-12  [id 240]

(in reply to Report 2 on 2018-03-15)

Errors in user-supplied markup (flagged; corrections coming soon)

Like as for the first referee, we would express our gratitude to the second referee for the careful reading of the manuscript and for bringing up valuable suggestions for improvements. A revised version (v2) of the manuscript has been submitted. Please see our replies below.

$\textbf{Point 1}$
==================================================
The first part of the paper describing the basis idea is clearly written. The construction of the Fock state basis using the natural orbitals could be described in more detail. The interpretation of the data in Fig. 1 is a bit unclear. For n=0, it is clear that the participation is tending to 1 in the localized phase. Why doesn’t the same effect occur for n>0? Why does n=1 curve for P^{(n)} appear qualitatively different from the n=0 curve? Even in the thermal phase n=3,4 curves are around 0.5. I would have expected to be smaller. The natural orbitals presumably in the thermal phase are completely delocalized. Is that true? Since, this is an important aspect of the work and probably the subsequent analysis depends crucially on the behaviour of this quantity, it would be suitable if the the data and its interpretation is clarified substantially. Presumably, further clarity can be achieved by studying the finite size scaling of this quantity.
==================================================

In the revised manuscript, we have elaborated on the construction of the Fock state basis and the interpretation of $P^{(n)}$ in more detail. As pointed out in the revised manuscript, the scaling of $P^{(n)}$ with $n$ depends on the system size in a non-trivial way, such that no simple finite-size scaling analysis is possible for this quantity. With the aim of giving a flavour of the dependence of $P^{(n)}$ on the system size, we have added a plot of $P^{(n)}$ as a function of $W$ for $L=14$ next to the one for $L=16$ in Figure 1 of the revised manuscript.

The question whether the natural orbitals are delocalized in the thermal phase has been studied in Ref. [23] of the revised manuscript. Even though the natural orbitals are far more extended in the thermal phase compared to the many-body localized phase, it is unclear whether this can be seen as true delocalization. We have added the above statement in the revised manuscript.

$\textbf{Point 2}$
==================================================
The average PR shown in fig. 2 shows the two different regimes. The exponential growth with system size at low disorder in the thermal and the convergence to 1 for all system sizes clearly show the two regimes quite clearly. The authors use this to conclude that the MBL regime sets in at lower disorder. This is likely correct but the analysis used to reach this conclusion is a bit loose. Scaling the data to extract the critical disorder strength would more thoroughly demonstrate this point. The authors should do this for further clarity.
==================================================

Please see our reply to point 1 of report 1.

$\textbf{Point 3}$
==================================================
The authors also use the variance of the participation to find the crossover regime. For system sizes 10 and 12, the peak is imperceptible but becomes stronger for system sizes 14 and 16. The delocalised eigenstates satisfy the eigenstate thermalization hypothesis. What is the scaling of the participation ratio expected in the thermal phase based on that? By using the appropriate scaling the location of the critical regime can be better estimated from this quantity.
==================================================

Unfortunately, we were not able to use scaling arguments to make a better estimate of the location of the crossover when focusing on the variance of the participation ratio. Figure 5 of the revised manuscript shows that the eigenstates acquire a non-trivial structure in a rather wide region around the crossover. As a consequence, scaling arguments based on the behaviour expected when the eigenstate thermalization hypothesis applies can not be used without inducing large uncertainties.

---

## Round 2 · Referee Report · Anonymous · 2018-5-24

Strengths

1- Interesting and timely subject
2- interesting approach to Fock space localization

Weaknesses

1- The authors addressed in their new version the comments of the referees. In
particular, they included more details on the construction of the Fock space,
although I think it would be beneficial for the reader to include a definition
of $|\psi_i^{(n)}\rangle$ in terms of an equation, to clarify that the creation and
annihilation operators $d_i^{(\dagger)}$ are used. Also a clearer definition of
$|\psi^{(0)}\rangle$ would be useful.

2- I think it should in general be expected that the quantity $P^{(n)}$ does not
show a sharp signature of the MBL transition, since Slater determinants in terms
of natural orbitals only become good approximations of eigenvectors in the limit
of large disorder, where one expects $P^{(0)}=1$.

3- Fig. 4 shows not the ratio of the typical PR with the dimension of the Hilbert
space, illustrating strong localization in the Fock basis at strong disorder.
The authors observe a crossing of curves corresponding to different system
sizes and interpret this as an indication for a lower critical disorder
(compared to $W_c\approx 3.6$ in the literature). However, this interpretation
ignores a very strong drift of the crossings of consecutive system sizes towards
higher disorder and I think this statement should therefore be removed. One can
not seriously conclude a smaller critical disorder for $L\to \infty$ from the presented data.

4- Minor remark: Just below Eq. (1), "commutator" should be changed to "anticommutator".

Report

1- The authors now added errorbars to all results based on a jackknife resampling,
showing that fluctuations in particular for $L=16$ in Fig. 3 are of statistical
nature and of the order of the size of the errorbars.

2- The Histograms for the $\log_{10}(PR)$ shown in Fig. 5 demonstrate that the peak in
the variance at intermediate disorder originates from broad distributions,
although the distributions are not bimodal. This is consistent with what is
observed e.g. in the entanglement entropy. Interestingly, the broadest
distributions seem to be at slightly different positions in the two bases (lower
disorder in the Fock basis). Maybe the authors could plot at least the L=16
result in Fig 3 for both bases in both panels for comparison.

Requested changes

1- Add explicit definitions of $|\psi_i^{(n)}\rangle$ in terms of equations.
2- Remove statement that the data suggests a lower value of $W_c$, discuss instead drifts with system size.

  • validity: good
  • significance: good
  • originality: high
  • clarity: good
  • formatting: excellent
  • grammar: excellent

Anonymous on 2018-05-31  [id 262]

(in reply to Report 1 on 2018-05-24)

We are grateful to the referee for the positive report and the useful suggestions to improve the manuscript. We reply to the points raised below in the order in which they appear in the report.

${\textbf{Point 1}}$

The authors addressed in their new version the comments of the referees. In particular, they included more details on the construction of the Fock space, although I think it would be beneficial for the reader to include a definition of $| \psi_i^{(n)} \rangle$ in terms of an equation, to clarify that the creation and annihilation operators $d_i^{(\dagger)}$ are used. Also a clearer definition of $|\psi^{(0)} \rangle$ would be useful.

In the revised manuscript, we have followed this suggestion.

${\textbf{Point 2}}$

I think it should in general be expected that the quantity $P^{(n)}$ does not show a sharp signature of the MBL transition, since Slater determinants in terms of natural orbitals only become good approximations of eigenvectors in the limit of large disorder, where one expects $P^{(0)} = 1$.

To avoid the suggestion that $P^{(n)}$ should show a sharp signature of the MBL transition, we have replaced the sentence "Interestingly, no clear $\ldots$ the MBL transition'' by "No clear signatures of the MBL transition can be observed, and on average eigenstates seem to remain localized at disorder strengths even below the MBL transition.'' in the revised manuscript. However, we would like to stress that the succesful use of the occupation discontinuity $ \Delta n$ as a probe for the MBL transition in $\it{e.g.}$ Ref. [23] suggests that the structure of eigenstates in the Fock basis changes qualitatively across the MBL transition, which one might expect to be reflected in $P^{(n)}$ in some (probably non-trivial) way.

${\textbf{Point 3}}$

Fig. 4 shows not the ratio of the typical PR with the dimension of the Hilbert space, illustrating strong localization in the Fock basis at strong disorder. The authors observe a crossing of curves corresponding to different system sizes and interpret this as an indication for a lower critical disorder (compared to $W_c \approx 3.6$ in the literature). However, this interpretation ignores a very strong drift of the crossings of consecutive system sizes towards higher disorder and I think this statement should therefore be removed. One can not seriously conclude a smaller critical disorder for $L \to \infty$ from the presented data.

We are not aiming to suggest that the critical disorder strength $W \approx 3.6$ is incorrect. In the revised manuscript, we have removed phrasings that might lead to this impression. Unofrtunately, we were not able to make conclusive statements about the $L \to \infty$ behaviour, which we mention in the revised manuscript.

${\textbf{Point 4}}$

Minor remark: Just below Eq. (1), ''commutator'' should be changed to ''anticommutator''.

We thank the referee for bringing this to our attention. We have fixed this in the revised manuscript.

---

## Round 2 · List of Changes

Besides the changes mentioned in the replies to the referee reports, we have made the following changes:

- We have re-organized Section 4 in view of the new content added.
- We have updated Ref. [26] of the revised manuscript from the arXiv to the published version, added Ref. [30] of the revised manuscript, and removed Ref. [21] of the original manuscript as the relevant content is also covered in Ref. [7] of the original manuscript.
- We made several minor changes in the text and graphics with the aim of increasing clarity and/or readibility.

---

## Round 3 · List of Changes

Besides the changes mentioned in the reply to the referee report, in Figure 3 we have duplicated the $L=16$ data to both the left and right panel.

---

## Editorial Decision

published